# Time trends in healthy lifestyle among adults in Germany: Results from three national health interview and examination surveys between 1990 and 2011

**Jonas D. Finger** *, **Markus A. Busch, Christin Heidemann, Cornelia Lange, Gert B. M. Mensink, Anja Schienkiewitz**

Department of Epidemiology and Health Monitoring, Robert Koch Institute, Berlin, Germany

\* FingerJ@rki.de

## Abstract

### Background

The combined impact of multiple healthy behaviors on health exceeds that of single behaviors. This study aimed to estimate trends in the prevalence of a healthy lifestyle among adults in Germany.

### Methods

A data set of 18,058 adults aged 25–69 years from three population-based national health examination surveys 1990–92, 1997–99 and 2008–11 with complete information for five healthy behavior factors was used. A 'daily intake of both fruits and vegetables, 'sufficient physical exercise', 'no current smoking' and 'no current risk drinking' were assessed with self-reports and 'normal body weight' was calculated based on measured body weight and height. A dichotomous 'healthy lifestyle' indicator was defined as meeting at least four out of five healthy behaviors. Age-standardized prevalence was calculated stratified by sex, age groups (25–34, 35–44, 45–54 and 55–69 years) and education level (low, medium and high). Trends were expressed in relative change (RC) between 1990–92 and 2008–11.

### Results

In Germany, the overall prevalence of healthy lifestyle increased from 9.3% in 1990–92 to 13.5% in 1997–99 and to 14.7% in 2008–11 (RC: +58.1%). The prevalence increased among men and women and in all age groups, with the exception of men aged 45–54 years. The RC of increasing healthy lifestyle prevalence between 1990–92 and 2008–11 was stronger albeit on a higher level among women compared to men. Therefore, the gender difference in healthy lifestyle has increased, but age-related differences have overall decreased in this period. Among high educated men the prevalence of a healthy lifestyle increased between 1990–92 and 2008–11 from 10.6% to 16.3% (p = 0.01) and among high educated women from 16.4% to 30.3% and also among medium educated women

**Data Availability Statement:** The data from the GNHIES studies cannot be made publicly available because informed consent from study participants

did not cover public deposition of data and publicly providing an anonymized version of the analytical data set used in our current analysis would not comply with current data protection regulations in Germany as anonymized information could still be used in combination and/or with other data to identify DEGS study participants. However, the minimal data set underlying the findings presented in this article is archived in the 'Health Monitoring' Research Data Centre at the Robert Koch Institute (RKI) and can be accessed by all interested researchers on site. The 'Health Monitoring' Research Data Centre is accredited by the German Data Forum according to uniform and transparent standards. On-site access to the minimal data set is possible at the Secure Data Center of the RKI´s 'Health Monitoring' Research Data Centre (e-mail: fdz@rki.de).

**Funding:** The authors received no specific funding for this work.

**Competing interests:** The authors have declared that no competing interests exist.

(10.9 to 16.6, p<0.01), but no significant increase in healthy lifestyle prevalence was observed among men with low and medium education and among women with low education level.

## Conclusions

The prevalence of a lifestyle with at least four out of five healthy behaviors markedly increased from 1990–92 to 2008–11. Nevertheless, additional health promotion interventions are needed to improve the number of combined healthy behavior factors and the awareness in the population that each additional healthy behavior factor leads to a further improvement in health, especially in men in the age-range 45 to 54 years, and among persons with low education level.

## Introduction

A healthy lifestyle is a combination of several healthy behaviors that reduce the likelihood of ill-health and premature death [1]. These usually include non-smoking, sufficient physical activity, healthy eating and low-risk alcohol drinking. Furthermore, normal body weight often is considered to be a healthy behavior related factor. Different approaches have been applied to combine several healthy behavior factors to composite indices of healthy lifestyle [2–5]. Studies using such measures demonstrated a clear dose-response relationship: the higher the number of combined healthy behavior factors, the larger are the health benefits [6–13]. A recent cohort study using NHANES data with a 35 years follow-up period observed a risk reduction of 74% for all-cause mortality among persons with five healthy behaviors compared to those without healthy behavior factors [14]. In addition, an unhealthy lifestyle with a lower number of healthy behavior factors has been linked to a higher risk of developing type 2 diabetes and cardiovascular diseases and to premature death [15–17]. Another recent study based on the European Social Survey 2014 observed that only 5.8% of adults in Europe were adhering to five healthy behaviors with strong differences between the 20 countries under study [18]. Hungary with 1.3% and the Czech Republic with 1.9% were the countries with the lowest prevalence of five healthy behaviors and the United Kingdom with 8.6% and Finland with 9.2% were those with the highest prevalence [18]. In Germany in 2009–10, 7.1% of women and 3.2% of men were adhering to five healthy behaviors and 29.1% of women and 17.8% of men showed at least four out of five healthy behaviors [19]. Studies estimating the potential of reducing social inequalities in mortality in Europe by simulating scenarios in which educational groups have the same healthy lifestyle pattern as the high-educated group, showed that the potential is very high for countries with a marked social gradient of healthy behaviors [20–22]. Thus, information on multiple healthy behavior factors combined is crucial for public health monitoring and policy planning. However, only a few studies have analyzed trends over time in healthy lifestyle in the general adult population and these observed only little net changes in the prevalence of healthy lifestyles [23].

For Germany, time trends of prevalence of a healthy lifestyle in the adult population have not been investigated so far. Therefore, the aim of this study is to estimate trends in the prevalence of a healthy lifestyle among adults in Germany according to gender, age and level of education.

## Materials and methods

### Study design, setting and participants

We used data of the German Federal Health Monitoring System from three cross-sectional German national health interview and examination surveys (GNHIESs) with information about health status, risk factors and health behavior for the general adult population [24]. The surveys were conducted from 1990 to 1992 (GNHIES 1990–92), 1997 to 1999 (GNHIES 1997–99) and 2008 to 2011 (GNHIES 2008–11) [25–27]. Detailed information on the study designs and methods for the three surveys were published previously [25–28]. In brief, all three surveys were based on two-stage cluster random sampling designs. One hundred twenty to 180 sample points (clusters) were randomly selected for each survey proportional to the population structure of the Federal Republic of Germany. In a second step, address information were randomly drawn from local population registries in the sampled communities. The eligible study populations were adults living in Germany aged 18 to 79 years for GNHIES 1997–99 and GNHIES 2008–11 and 25- to 69-year-old adults with German nationality for GNHIES 1990–92. Hence, we restricted our trend analysis to the age range 25 to 69 years [28]. The response rates were 70% for GNHIES 1990–92, 61% for GNHIES 1997–99 and 42% for GNHIES 2008–11 [28]. A non-responder analysis for GNHIES 2008–11 indicated a high representativeness between the study sample and the German population structure [29]. All participants were informed about the study objectives, examination and interview processes and applicable data protection guidelines. Verbal consent was witnessed and formally recorded in the surveys 1990–92 and 1997–1999. The GNHIES 1997–99 and GNHIES 2008–11 participants signed an informed written consent prior to participation. All surveys were conducted according to the Federal and State Commissioners for Data Protection guidelines and the GNHIES 2008–11 study protocol was approved by the Charité –University Medicine Berlin ethics committee; No. EA2/047/08 [27]. The 1997–99 and 2008–11 13 surveys conform to the principles of the Helsinki Declaration.

### Measurements and variables

In line with previous studies [13, 14, 19], we selected five healthy behavior factors to construct a healthy lifestyle index (HLI)–sufficient physical activity, low-risk alcohol drinking, a healthy diet (fruit and vegetable intake), non-smoking, and normal body mass index (BMI). Appropriate data with comparable assessment methods across the three surveys were identified for each healthy behavior factor. Physical exercise and smoking data were assessed with self-administered questionnaires as described elsewhere [28]. Briefly, physical exercise was assessed with the question 'How often do you engage in physical exercise?' and the regular duration in hours per week was assessed with categorical answer options. Smoking habits were assessed with the questions allowing a distinction between 'current', 'former' and 'never' smoking [6, 9]. In all surveys, alcohol as well as fruit and vegetable consumption were assessed with self-administered food-frequency questionnaires; however, there are differences in the data collection across the surveys in food groups, reference time, intake frequencies and additional information on portion sizes. To compare information on alcohol consumption across surveys we used frequencies and quantities of alcoholic beverage intake. A detailed description of the alcohol assessment is given in the supplement (S1 File).

A daily consumption of both fruits and vegetables was used as an indicator for a healthy dietary pattern. To compare the information on fruit and vegetable consumption across the three surveys we used only frequency information since portion sizes were not obtained in GNHIES 1990–92 and GNHIES 1997–99. Information on frequency of consumption of

fruits and vegetables was assessed in GNHIES 1990–92 with the question 'How often do you consume these particular foods?' without a specific reference period. Participants reported their consumption of 'cooked vegetables', 'tinned vegetables', 'salad', 'raw vegetables', and 'fresh fruits' using the frequency categories '(almost) daily', 'several times a week', 'about once a week', '2 to 3 times a month', 'once a month or less', and 'never'. In GNHIES 1997–99, data on frequency of intake during the past 12 months was collected for 'cooked vegetables', 'tinned vegetables', 'lettuce, raw salad, raw vegetables', and 'fresh fruits' using the frequency categories 'several times a day', 'daily or almost daily', 'several times a week', 'about once a week', '2 to 3 times a month', 'once a month or less', and 'never'. In GNHIES 2008–11, the frequency of intake during the last 4 weeks was assessed for ' raw vegetables', 'cooked vegetables', 'legumes', 'fresh fruits' and 'cooked fruits'. The answering categories were 'more than 5 times a day', '4 to 5 times a day', '3 times a day', 'twice a day', 'once a day', '5 to 6 times a week', '3 to 4 times a week', '1 to 2 times a week', ' 2 to 3 times a month, 'once a month or less', and 'never'. The dietary assessment method in GNHIES 2008–11 has been described in detail previously [30]. To examine trends over time from surveys with different dietary collection methods, we standardized the frequency of fruit and vegetable consumption by recoding the information into a dichotomous variable reflecting a daily intake of both fruits and vegetables or not.

In all surveys, body weight and body height were measured during physical examination in a standardized manner by trained personal. BMI was calculated as the ratio of a person's body weight to the square of body height ($kg/m^2$).

**Outcome variables.** Smoking status was categorized into 'current smoking' versus 'no current smoking'. This cut-off was selected because it was used in healthy lifestyle scores of previous studies which demonstrated significant associations between healthy lifestyle and health outcomes and mortality [6, 9]. 'Sufficient physical exercise' was defined as reporting regular physical exercise of at least two hours per week. This cut-off point was selected because from the available answer options it comes closest to the minimum level of health-enhancing physical activity of 2.5 hours per week recommended by the World Health Organization (WHO) [31]. Moreover, in a previous study we demonstrated that this level of physical exercise is associated with an improved cognitive function across the life span [32]. 'No current risk drinking' was defined as consuming $\leq 20$ grams of pure alcohol per day in men and $\leq 10$ grams per day in women in line with evidence-based drinking guidelines for German adults [33, 34]. 'Daily fruits and vegetables' intake was defined as once or more per day versus less than daily intake of fruits and vegetables. 'Normal weight' was defined according to WHO guidelines as having a BMI in the range between 18.5 to less than 25 $kg/m^2$ [35]. A 'Healthy lifestyle index' (HLI) was constructed by assigning one point for each healthy behavior factor resulting in a score from zero to five; where zero is interpreted as high-risk and five as low-risk lifestyle. Furthermore, a dichotomous variable with a cut-off of 'at least four' healthy behavior factors (HLI $\geq 4$) was used and hereafter referred to as 'healthy lifestyle'. The selection of this cut-off was based on a meta-analysis which indicated that a combination of at least four healthy behavior factors is associated with a reduction of all-cause mortality by 66% [13].

**Stratification variables.** We stratified the analyses by gender and used the following age strata: 25 to 34, 35 to 44, 45 to 54 and 55 to 69 years. 'Level of education' was defined according to the International Standard Classification of Education (ISCED) 2011 and EUROSTAT guidelines [36]. Based on self-reported information on the highest education level, education of participants was classified as low (ISCED levels 0–2), medium (ISCED levels 3–4) or high education (ISCED levels 5–8) [36].

**Table 1. Distribution of the study samples according to sex and age.**

| | GNHIES 1990–92 | GNHIES 1997–99 | GNHIES 2008–11 | Total |
|---|---|---|---|---|
| | 1990–92 | 1997–99 | 2008–11 | |
| | n (%) | n (%) | n (%) | n |
| Total sample | 7466 | 5825 | 5375 | 18666 |
| Study sample | 7382 | 5603 | 5073 | 18058 |
| Men | 3594 (48.7) | 2742 (48.9) | 2380 (46.9) | 8716 |
| Age | | | | |
| 25–34 | 922 (25.7) | 614 (22.4) | 378 (15.9) | 1914 |
| 35–44 | 839 (23.3) | 698 (25.5) | 455 (19.3) | 1992 |
| 45–54 | 905 (25.2) | 578 (21.1) | 634 (26.6) | 2117 |
| 55–69 | 928 (25.8) | 852 (31.1) | 913 (38.4) | 2693 |
| Women | 3788 (51.3) | 2861 (51.1) | 2693 (53.1) | 9342 |
| Age | | | | |
| 25–34 | 1022 (27.0) | 646 (22.6) | 410 (15.2) | 2078 |
| 35–44 | 856 (22.6) | 720 (25.2) | 536 (19.9) | 2112 |
| 45–54 | 883 (23.3) | 587 (20.5) | 755 (28.0) | 2225 |
| 55–69 | 1027 (27.1) | 908 (31.7) | 992 (36.8) | 2927 |

## Study size

The total numbers of participants in the age group 25 to 69 years were 7,466 for GNHIES 1990–92 (3,641 men and 3,825 women), 5,825 for GNHIES 1997–99 (2,831 men and 2,994 women) and 5,375 for GNHIES 2008–11 (2,538 men and 2,837 women). After exclusion of individuals with missing data for at least one healthy behavior factor used for the HLI, the final study sample consisted of 7,382 participants for GNHIES 1990–92, 5,603 for GNHIES 1997–99 and 5,073 for GNHIES 2008–11 (Table 1). Those numbers relate to item response rates for the HLI of 98.9% for the GNHIES 1990–92, 96.2% for the GNHIES 1997–99 and 94.4% for the GNHIES 2008–11. The total study sample comprised 18,058 participants, 8,716 men and 9,342 women.

## Statistical methods

We used the software SAS Version 9.4 (SAS Institute, Cary, NC, USA) for the statistical analyses. Within the analyses, a weighting procedure was applied to all three surveys to adjust for deviations from the German population structure at each survey period according to age, sex, region and education level. In addition, the results were age-standardized to the German population structure as of 31 December 2010 to control for demographic changes and differences in age distribution across the three survey samples. Trend estimations were performed, while adjusting for survey design effects of the cluster sampling designs of each survey. Age-standardized and weighted prevalence and 95% confidence intervals (CI) were calculated for each healthy behavior factor separately and for the HLI based on three cross-sectional survey samples collected at three different time periods (1990–92, 1997–99 and 2008–11). Trends were expressed with relative change (RC), i.e. (value in survey 3 minus value in survey 1) / (value in survey 1) × 100%. The basis for comparison is always the estimate from the first survey (GNHIES 1990–92). Logistic regression was used to test time trends for statistical significance with the survey wave as a categorical variable in the model. The criterion for statistical significance was set at $p < 0.05$ [28].

## Results

### Healthy behavior factors

Time trends for the five healthy behavior factors are presented for men and women separately in Fig 1. Among men and women, the prevalence of sufficient physical exercise and no current risk drinking have increased between 1990–92 and 2008–11 (RC: sufficient physical exercise +44.9% in men; +118.2% in women); no current risk drinking +64.8% in men and +53.1% in women; all p<0.01), while daily fruits and vegetables intake has decreased (RC: -38.6% in men and -29.2% in women; both p<0.01). More pronounced differences were observed for no current risk drinking with higher increases in prevalences between 1990–92 and 1997–99 compared to 1997–99 and 2008–11. The decrease in daily fruits and vegetables intake was more distinct between 1997–99 and 2008–11 in comparison to 1990–92 and 1997–99. The prevalence of no current smoking has increased among men (+7.2%, p = 0.04) but decreased among women (-4.3%, p = 0.10). Between 1990–92 and 2008–11 no statistical relevant changes can be observed among men and women for the prevalence of normal weight. In all age groups sufficient physical exercise and no current risk drinking increased as well as daily fruits and vegetables intake decreased over time (S1 and S2 Tables). No current smoking increased over time among men in the age groups 35–44 years (RC: +17.9%, p<0.01) and 55–69 years (RC: +13.7%, p<0.01) but decreased in 45–69 year old women (45–54 years RC: -12.7%, 55–69 years RC: -8.2%, both p<0.01). A higher prevalence of normal weight in 2008–11 compared to 1990–92 was only observed among older women (45–54 years RC: +14.5%, p = 0.08; 55–69 years RC: +26.3%, p<0.01). The increasing trend in sufficient physical exercise and no current risk drinking and the decreasing trend in daily fruits and vegetables intake over time can be

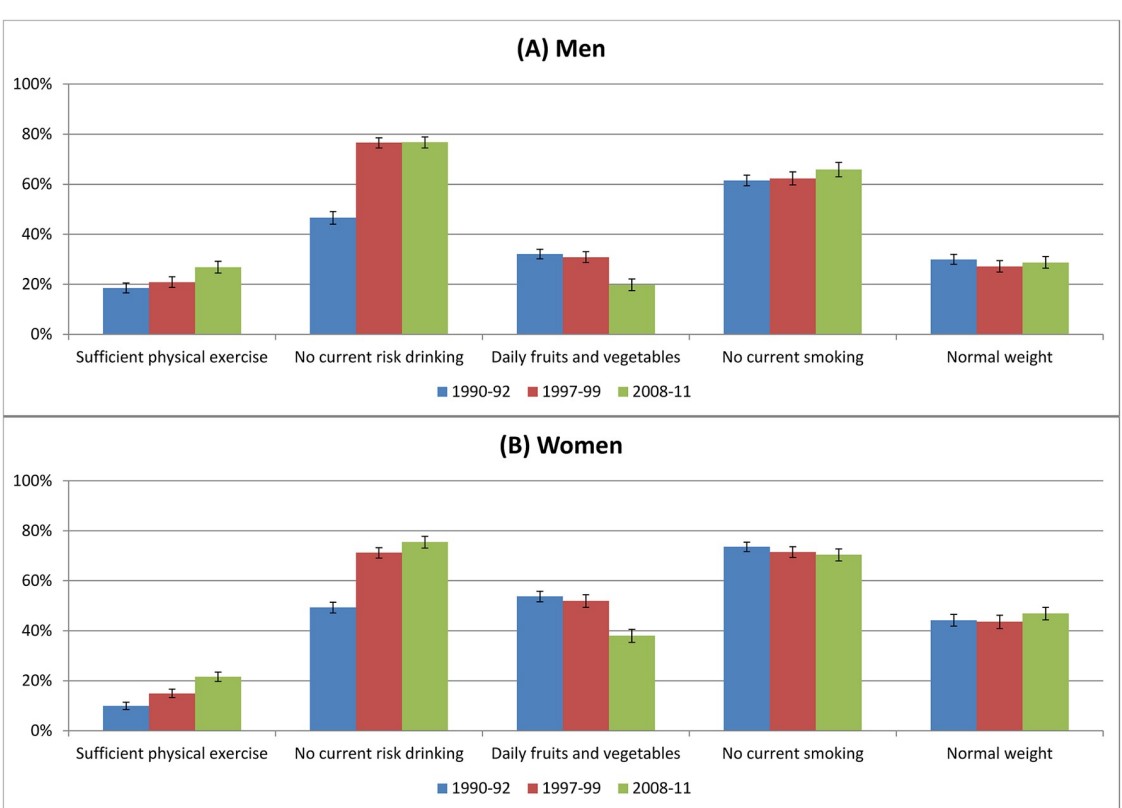

**Fig 1. Prevalence (%, 95%-CI) of individual healthy behavior factors among men (A) and women (B) aged 25–69 years.**

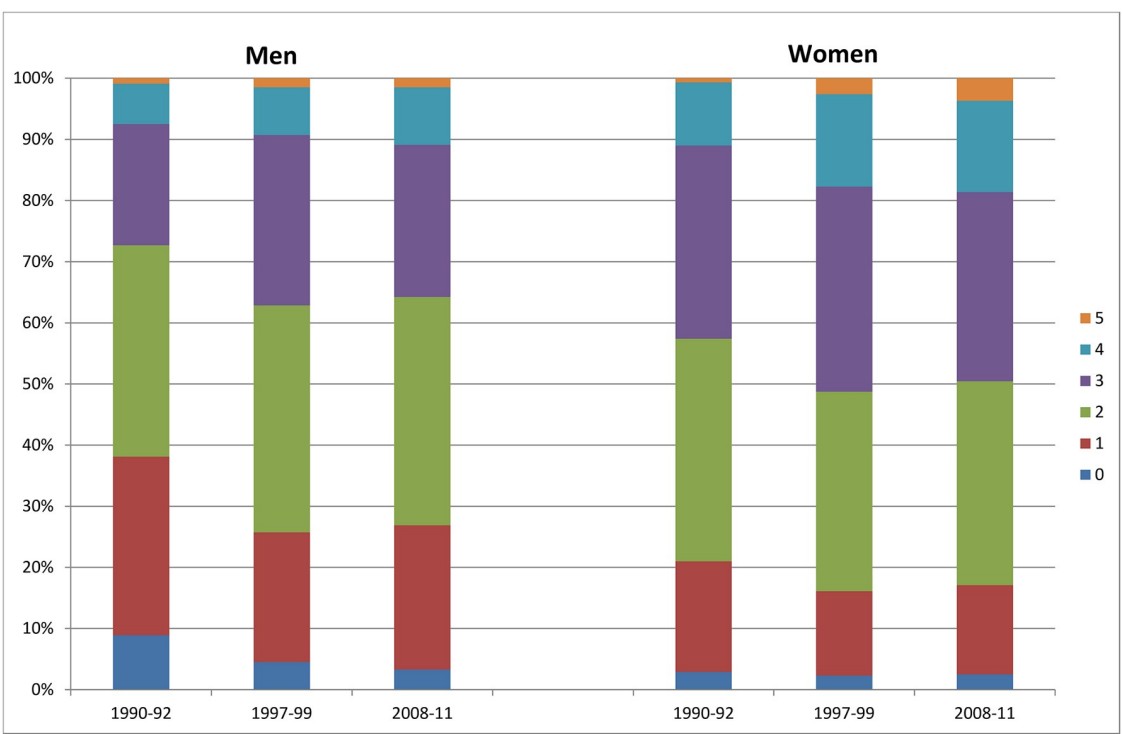

**Fig 2. Proportions of specific numbers of healthy behavior factors among men and women aged 25–69 years.**

observed in all educational groups albeit on a different level (S1 Fig). No significant changes in the prevalence of normal weight over time between different educated groups can be observed.

## Healthy lifestyle index (HLI)

In total, the proportion with a high number of combined healthy behavior factors has increased between 1990–92 and 2008–11 (Fig 2). The proportions of men with none or only one healthy behavior factor have declined (relative change: -62.9% and -19.2%, both p < 0.0001), while the proportions of those with three (RC: +25.8%, p = 0.10), four (RC: +42.4%, p = 0.009) and five factors (RC: +66.7%, p < 0.0001) have increased. Among women, the proportion of those with four and five healthy behavior factors has strongly increased (RC: +44.7% and +428.6, both p<0.0001), while the proportions of those with less than 4 factors have declined. However, this result is only statistically significant among women with one (RC: -19.3%, p = 0.0004) or two (RC: -8.5%, p = 0.02) healthy behavior factors. In general, among all age groups, the proportions of men and women with none and one healthy behavior factors have declined between 1990–92 and 2008–11, while the proportions of adults with four and five factors have increased (Fig 3). However, not all trends are statistically significant.

## Healthy lifestyle (HLI ≥ 4)

Table 2 presents the proportions of each health behavior factor by HLI ≥ 4. The proportion of sufficient physical exercise, daily fruits and vegetables, no current smoking, and normal weight as proportion of HLI ≥ 4 increased over time. No current risk drinking did not show any changes over time.

The proportions of adults with a healthy lifestyle across the surveys are presented in Table 3. HLI ≥ 4 prevalence increased in total from 9.3% in 1990–92 to 13.5% in 1997–99 and

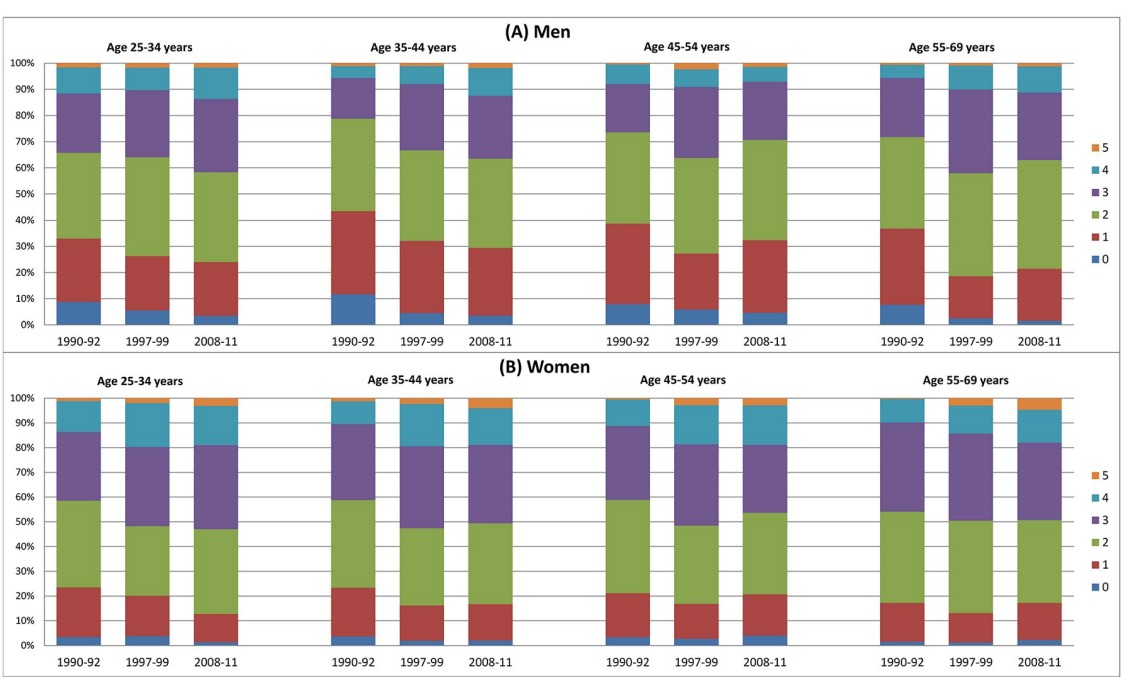

**Fig 3. Proportions of specific numbers of healthy behavior factors among men (A) and women (B) by age groups.**

to 14.7% in 2008–11 (RC 1990-92-2008-11: +58.1%, p<0.01). The corresponding percentages were for men 7.5%, 9.3% and 10.9% (RC: +45.3%, p<0.01) and for women 11.1%, 17.7% and 18.6% (RC: +67.6%, p<0.01). The prevalence increased in all age groups, with the exception of men aged 45–54 years. The relative change of increasing healthy lifestyle prevalence between 1990–92 and 2008–11 was stronger albeit on a higher level among women compared to men. Therefore, gender difference in healthy lifestyle has increased, but age-related differences have overall decreased in this period.

Some sex differences in the trends for age and education were observed. While in women the proportion with a healthy lifestyle increased between 1990–92 and 2008–11 in all age groups, among men increases were only observed in the age groups 35 to 44 years and 55 to 69 years. The proportion of women with a healthy lifestyle increased between 1990–92 and 2008–11 only in the high and medium education groups, while among men an increase was only observed in the high education group.

## Discussion

In this trend analysis, based on three population-based German national health interview and examination surveys, it is observed that the prevalence of a healthy lifestyle defined as having

**Table 2. Proportion (%, 95%-CI) of each health behaviour factor by healthy lifestyle index (HLI) ≥ 4.**

|  | HLI ≥ 4 | | |
|---|---|---|---|
|  | **1990–92** | **1997–99** | **2008–11** |
| Sufficient physical exercise | 33.9 (30.9–37.1) | 45.0 (41.6–48.4) | 42.7 (39.1–46.3) |
| No current risk drinking | 16.5 (14.8–18.3) | 17.2 (15.7–18.9) | 18.3 (16.7–20.1) |
| Daily fruits and vegetables | 19.2 (17.4–21.2) | 28.0 (25.6–30.6) | 38.6 (35.6–41.7) |
| No current smoking | 13.3 (11.9–14.9) | 19.4 (17.7–21.2) | 21.0 (19.1–22.9) |
| Normal weight | 21.6 (19.6–23.7) | 30.3 (27.7–32.9) | 31.5 (28.6–34.5) |

**Table 3. Proportions (%) of adhering to a healthy lifestyle[1] among men and women according to sex, age group and education.**

|  | 1990–92 | 1997–99 | 2008–11 | Relative change | p[#] |
|---|---|---|---|---|---|
|  |  |  |  | 1990–92 to 2008–11 |  |
|  | % (95% CI) | % (95% CI) | % (95% CI) | % |  |
| **Total** | 9.3 (8.3–10.3) | 13.5 (12.3–14.7) | 14.7 (13.4–16.2) | 58.1 | < .0001 |
| Age |  |  |  |  |  |
| 25–34 | 12.6 (11.0–14.4) | 14.9 (12.8–17.2) | 16.3 (13.3–19.7) | 29.4 | 0.08 |
| 35–44 | 8.1 (6.6–9.8) | 13.5 (11.8–15.5) | 15.6 (13.1–18.6) | 92.6 | < .0001 |
| 45–54 | 9.5 (8.0–11.3) | 13.8 (11.6–16.4) | 12.9 (10.9–15.2) | 35.8 | 0.005 |
| 55–69 | 7.8 (6.4–9.5) | 12.2 (10.4–14.3) | 14.7 (12.3–17.4) | 88.5 | < .0001 |
| Education |  |  |  |  |  |
| Low | 7.3 (5.9–9.1) | 8.3 (6.5–10.4) | 9.0 (6.4–12.7) | 23.3 | 0.57 |
| Medium | 8.8 (7.8–10.0) | 12.8 (11.6–14.2) | 12.6 (11.0–14.3) | 43.2 | < .0001 |
| High | 12.5 (10.8–14.5) | 19.7 (17.3–22.3) | 22.2 (19.8–24.8) | 77.6 | < .0001 |
| **Men** | 7.5 (6.4–8.8) | 9.3 (8.1–10.8) | 10.9 (9.4–12.5) | 45.3 | 0.003 |
| Age |  |  |  |  |  |
| 25–34 | 11.6 (9.5–14.1) | 10.3 (8.0–13.2) | 13.6 (10.0–18.2) | 17.2 | 0.40 |
| 35–44 | 5.7 (4.1–8.0) | 8.0 (5.9–10.7) | 12.4 (9.1–16.8) | 117.5 | 0.004 |
| 45–54 | 7.9 (6.0–10.4) | 9.1 (6.7–12.2) | 7.1 (5.0–9.9) | -10.1 | 0.61 |
| 55–69 | 5.7 (4.1–7.8) | 10.1 (7.8–12.8) | 11.3 (8.7–14.5) | 98.2 | 0.003 |
| Education |  |  |  |  |  |
| Low | 3.4 (1.9–6.0) | 4.0 (2.0–7.8) | 7.6 (3.9–14.3) | 123.5 | 0.19 |
| Medium | 6.9 (5.8–8.2) | 8.2 (6.7–9.9) | 8.5 (6.9–10.5) | 23.2 | 0.29 |
| High | 10.6 (8.4–13.2) | 13.9 (11.5–16.8) | 16.3 (13.5–19.4) | 53.8 | 0.01 |
| **Women** | 11.1 (9.9–12.4) | 17.7 (15.9–19.7) | 18.6 (16.6–20.8) | 67.6 | < .0001 |
| Age |  |  |  |  |  |
| 25–34 | 13.6 (11.3–16.3) | 19.6 (16.5–23.3) | 19.0 (14.8–24.0) | 39.7 | 0.01 |
| 35–44 | 10.5 (8.4–13.1) | 19.4 (16.4–22.8) | 18.9 (15.4–23.1) | 80.0 | < .0001 |
| 45–54 | 11.2 (9.0–13.7) | 18.7 (15.2–22.7) | 18.9 (15.6–22.6) | 68.8 | 0.0003 |
| 55–69 | 9.8 (7.8–12.2) | 14.3 (11.7–17.4) | 18.0 (14.6–21.9) | 83.7 | 0.0005 |
| Education |  |  |  |  |  |
| Low | 9.0 (7.2–11.2) | 10.5 (8.2–13.5) | 10.0 (6.7–14.6) | 11.1 | 0.65 |
| Medium | 10.9 (9.2–12.8) | 17.6 (15.6–19.9) | 16.6 (14.1–19.3) | 52.3 | < .0001 |
| High | 16.4 (13.5–19.9) | 30.4 (25.9–35.2) | 30.3 (26.3–34.6) | 84.8 | < .0001 |

[#] p for trend

[1] Healthy lifestyle index ≥ 4

at least four out of five healthy behavior factors, overall increased explicitly among adults in the period between 1990–92 and 1997–99 and further slightly between 1997–99 and 2008–11. Age-related differences that were observed in 1990–92 with younger persons having a healthy lifestyle more often than older persons attenuated over time until 2008–11, among men and women. Gender-related differences that were observed in 1990–92 with women having more often a healthy lifestyle than men became slightly larger over time. Educational differences that were observed in 1990–92 showing that those with high education had a healthy lifestyle more often than those with low education became larger among women but not among men.

## Health implications

Cohort studies consistently demonstrated that the relative risk for all-cause and cardiovascular mortality proportionally decreased with a higher number of combined healthy behavior

factors [6, 12, 13]. Persons with a combination of at least four healthy behavior factors, as chosen in this trend study, can expect a reduction of all-cause mortality by 66% according to a meta-analysis including 15 cohort studies with more than 513,000 participants [13]. Hence, since the prevalence of a combination of at least four healthy behavior factors has increased between 1990–92 and 2008–11 in Germany, it can be expected that morbidity and mortality related with unhealthy behavior will subsequently decrease. In line with this assumption, the cardiovascular disease mortality has declined in Germany in the last decades and mortality rates are expected to further decline in Germany until 2025 [37–39]. This is on the one hand due to a better health care with improved detection and treatment of cardiovascular and metabolic diseases and on the other hand due to an improved health-related behavior [28]. Moreover, a previous trend study based on the same surveys indicated that mean systolic blood pressure, total cholesterol and serum glucose levels significantly declined in the period between 1990–92 and 2008–10 among men and women [28]. Those reductions in cardio-metabolic risk profiles and deaths may be partially explained by the observed increase in combined healthy behavior factors among adults in Germany. However, it is likely that an improved health care also contributed to the described developments with higher detection rates of undiagnosed hypertension, hyperlipidemia and diabetes and higher prescription and use of antihypertensive, lipid-lowering and antidiabetic drugs [28, 40–42].

## Trend patterns

To the best of our knowledge, trend studies on combined healthy behavior factors in adults using population-based, country-wide data are rare. A study from the USA observed only modest improvements in healthy lifestyle in adults in the period between 1994 and 2007 using data of the Behavioral Risk Factor Surveillance System [23]. Comparing single healthy behavior changes, non-smoking prevalence increased in the USA, while there was little change in fruit and vegetable consumption and physical activity, and prevalence of healthy weight even decreased [23]. The main drivers for increasing healthy lifestyle prevalence in our study were increases in sufficient physical exercise and no current risk drinking. In contrast daily fruits and vegetables intake decreased over time, which is also observed in the German National Nutrition Monitoring [43]. Yet, the largest part of observed improvements in single healthy behavior factors as well as in the HLI ≥ 4 occurred between 1990–92 and 1997–99 In Germany in the 1990s there was a discussion on a stronger legal alcohol limit when driving, which resulted in a revised law in 2001 [44]. Furthermore, the national legislation has developed the banning of alcohol use at work within the labor protection law and employer agreements at the workplace [45].

## Gender effects

The prevalence of a healthy lifestyle was significantly higher in women compared to men in all three observation periods. The relative change of increasing healthy lifestyle prevalence between 1990–92 and 2008–11 was stronger albeit on a higher level among women compared to men, leading to an increasing gender differences in healthy lifestyle over time. The strongest increases among all single healthy behavior factors were observed to be sufficient physical exercise in women and no current risk drinking in men. We are not aware of any other trend study to compare results on developments of gender-related differences in healthy lifestyle over time. However, Walther et al. also observed that female sex was a predictor for a higher healthy lifestyle score in a cohort of Swiss adults [46]. Several other studies using healthy lifestyle indices from the USA, Asia and Europe consistently showed that women are more likely to have a healthier lifestyle compared to men [6, 47–49].

## Age effects

In the observation period 1990–92, younger persons had overall higher healthy lifestyle prevalence than older persons, with a more pronounced pattern in men. Over time, healthy lifestyle prevalence increased in most age groups with the exception of the age groups 25–34 and 45–55 years among men and a less pronounced pattern in the age group 25–34 years among women leading to a less pronounced age difference in healthy lifestyle in 2008–11 compared to 1990–92. Among women, age differences in healthy lifestyle disappeared, with a prevalence of 18–19% in all age groups in 2008–11. Among men, the age group 45–55 years has the lowest healthy lifestyle prevalence at 7% and may therefore be a target group for health promotion interventions. The study sample consists of different generations including birth cohorts ranking from 1921 to 1985. People aged 55–69 years in 1990–92 belong to a birth cohort born between 1921/23–1935/37 and are referred to by social scientist as the 'Silent Generation', while people aged 55–69 years in 2008–10 belong to a birth cohort born between 1939/41–1953/55 of which the most belong to the 'Baby Boomers' generation [50]. It is likely that sociocultural differences between the generations exist which influences their lifestyles. The strong increase in sufficient physical exercise in the oldest age group 55–69 years is an important contributor to the positive trend in healthy lifestyle in this age group, especially among women. In the Baby Boomer generation in the course of the second wave feminist movement of the late 1960s and early 1970s [51] women adopted lifestyles which were formerly more common among men such as performing sports and exercise [52–54]. This may partly explain the higher prevalence of physical exercise in the Baby Boomer generation compared to the Silence Generation. In line with this observation, a trend study from Spain observed the same pattern that older people showed initially lower leisure-time physical activity levels than younger people and that this difference became smaller over time especially among women [55].

## Education effects

The prevalence of a healthy lifestyle was significantly higher among men and women with high compared to low education level in all three observation periods, this observation is in line with the previous finding from the 'German Health Update' study 2009 (GEDA 2009) [56]. An important driver for the overall increasing healthy lifestyle prevalence between 1990–92 and 2008–11 could be the educational expansion that took place in Germany in the last decades [57]. Studies simulated scenarios in which the low-educated adapt the health behaviors of the high-educated showing that societies can expect significant improvements in health equality [20–22]. This could partly have happened in Germany between 1990–92 and 2008–11 when the societal group of people with low education became smaller and the group of people with high education became larger over time. The educational expansion was driven by an disproportionate increase of females achieving high education certificates [58], leading to an improvement of gender equality in education. This is in line with the observation that healthy lifestyle prevalence has more strongly increased in females than in males. However, the educational expansion cannot explain differences in healthy lifestyle between educational groups. Among women, those with medium and high education showed stronger increases in healthy lifestyle over time than those with low education. This was different among men, where those with low education showed stronger increases in healthy lifestyle over time than those with medium and high education. However, only among men with high education the increase in healthy lifestyle prevalence was statistically significant. The result of increasing educational inequalities in healthy lifestyle among women is in line with previous trend studies showing

increasing educational inequalities over time in smoking and physical exercise behaviors among adults in Germany [59, 60].

## Limitations

Several limitations should be considered when interpreting our findings. All healthy behavior factors, except normal body weight, were based on self-reported information and thus, we cannot exclude the possibility that reporting bias occurred, e.g. because of limited cognitive ability to report complex behaviors or socially desirable answering [61, 62]. In addition, the surveys were conducted over a period of more than 20 years bringing about inevitably changes in survey and assessment methods making comparisons over time more difficult. We have paid much attention to these aspects when selecting variables and defining indicators, but we cannot fully exclude the possibility that methodological differences between the surveys have influenced the results [28, 63]. The initial physical exercise question has not changed across the three surveys, but the answer categories have been slightly adapted. The initial smoking question was the same in the first two surveys but was adapted in the third survey. The BMI assessment method based on physical examination data on body weight and height has remained consistent across the three surveys [28]. A consumption of both fruits and vegetables each day is a very rough indicator for a general healthy diet which may include much more aspects e.g. intake of highly processed foods, saturated fat, fiber, sugar intake. However, comprehensive information on the diet was not available in all three surveys. Consumption was assessed with self-administered food-frequency questionnaires with inconsistent reference periods and slightly different food items and answering categories for the three surveys. Although we tried to standardize this by condensing this information, this may have resulted in systematic differences in estimates for the surveys. Furthermore, we cannot fully rule out the possibility that selection bias has compromised generalizability of the results. As in many other European countries, the response rates in the German national health surveys have declined over time [64, 65]. The selection bias can be occurred at different levels of recruitment, e.g. selection of sample points, selection of individuals, or participation of individuals. However, we used weighting factors with a common methodology for all three surveys to increase the representativeness of the results [28].

## Conclusion

We conclude that prevalence of a lifestyle with at least four out of five healthy behaviors markedly increased between 1990–92 and 2008–11. Another positive development from a public health perspective is that age-related differences in healthy lifestyle have overall decreased during this period. On the negative side, the gender difference in healthy lifestyle prevalence has increased. Moreover, in men, increases in healthy lifestyle prevalence were limited to men aged 35 to 44 and 55 to 69 years and to those with high education, whereas in women, increases were observed in all age groups but limited to those with high or medium education. The awareness of a healthy lifestyle and the fact that each additional healthy behavior factor leads to a further improvement in health should be increased in the population. Additional health promotion interventions are needed to improve the number of combined healthy behavior factors in the general adult population. Special effort should be undertaken to reach men, especially in the age-range 45 to 54 years, and persons with low education level. According to the WHO, multi-component interventions should preferably be used to promote 'health in all policies' following a setting-specific approach and focusing on the upstream drivers of healthy behaviors in order to create more healthy environments, systems, societies and people [1, 66, 67].

## Supporting information

**S1 Fig. Proportions (%, 95%-CI) of healthy behaviour factors according to educational level.**
(TIFF)

**S1 Table. Proportions of individual healthy behavior factors among men aged 25–69 years according to age.**
(PDF)

**S2 Table. Proportions of individual healthy behavior factors among women aged 25–69 years according to age.**
(PDF)

**S1 File. Assessment of alcohol consumption within the German Federal Health Monitoring System.**
(PDF)

## Author Contributions

**Formal analysis:** Anja Schienkiewitz.

**Writing – original draft:** Jonas D. Finger.

**Writing – review & editing:** Markus A. Busch, Christin Heidemann, Cornelia Lange, Gert B. M. Mensink, Anja Schienkiewitz.

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
