## [Decision Letter · Decision Letter 0]

5 Jul 2019

PONE-D-19-16785

Time trends in healthy lifestyle among adults in Germany: Results from three national health interview and examination surveys between 1990 and 2011

PLOS ONE

Dear Dr Finger,

Thank you for submitting your manuscript to PLOS ONE. After careful consideration, we feel that it has merit but does not fully meet PLOS ONE’s publication criteria as it currently stands. Therefore, we invite you to submit a revised version of the manuscript that addresses the points raised during the review process.

We would appreciate receiving your revised manuscript by Aug 19 2019 11:59PM. To enhance the reproducibility of your results, we recommend that if applicable you deposit your laboratory protocols in protocols.io, where a protocol can be assigned its own identifier (DOI) such that it can be cited independently in the future. For instructions see: http://journals.plos.org/plosone/s/submission-guidelines#loc-laboratory-protocols

We look forward to receiving your revised manuscript.

Kind regards,

David Meyre

Academic Editor

PLOS ONE

Journal Requirements:

1.

2.

We note that you have indicated that data from this study are available upon request. PLOS only allows data to be available upon request if there are legal or ethical restrictions on sharing data publicly. For more information on unacceptable data access restrictions, please see http://journals.plos.org/plosone/s/data-availability#loc-unacceptable-data-access-restrictions.

Reviewers' comments:

Reviewer's Responses to Questions

**Comments to the Author**

1. Is the manuscript technically sound, and do the data support the conclusions?

Reviewer #1: Yes

Reviewer #2: Yes

Reviewer #3: Yes

2. Has the statistical analysis been performed appropriately and rigorously? 

Reviewer #1: Yes

Reviewer #2: I Don't Know

Reviewer #3: Yes

3. Have the authors made all data underlying the findings in their manuscript fully available?

Reviewer #1: No

Reviewer #2: Yes

Reviewer #3: Yes

4. Is the manuscript presented in an intelligible fashion and written in standard English?

Reviewer #1: Yes

Reviewer #2: Yes

Reviewer #3: Yes

5. Review Comments to the Author

Reviewer #1: Review MS number PONE-D-19-16785

Time trends in healthy lifestyle among adults in Germany: Results from three national

health interview and examination surveys between 1990 and 2011

Abstract

The abstract is well written and clear.

Methods: Please add e.g. sufficient/ xx amount to ‘Daily fruits and vegetables intake’.

Results: ‘Nevertheless, the gender difference in healthy lifestyle has increased’; Please clarify to which direction. Are men healthier compared to women?

Last sentence; start the sentence with referring to education, so it is directly clear that results for differences in education will be presented.

Conclusions: four out of five vs. 4 or 5 mentioned in the method section.

Ethics statement

Did the 1990-92 participants signed an informed consent?

Introduction

Line 73-77: Can authors also relate these numbers to a higher prevalence of health related diseases or premature death?

Materials and Methods

Line 92-94: Make clear that you compare prevalence data measured at 3 different time points in 3 different cohort. This is not always clear.

Line 104: Mentioning the response rate is not in line with informing that response rates for the three surveys were published previously (line 97).

Line 108 (as mentioned at the ethics statement): Why did participants from the 1997-99 wave did not sign an informed consent?

Line 119: ‘Smoking habits were assessed with the questions allowing a distinction between ‘current smokers’ and ‘others’’. This is a strict cut-off value. Please elaborate why this was so strict. Do authors have any information about former smoking?

Line 114 vs 121: clarify that a healthy diet is defined as fruit and vegetables intake.

Line 123-147: The explanation of alcohol consumption is too wordy which makes it difficult to follow. Recommend to rewrite/shorten or put to supplement.

Line 150: Assessment of fruit and vegetables in 1990-92 is not mentioned to which time period is referred to.

Line 148-168: Restructure, so all information is mentioned and provided in the same order.

Line 170 is repetitive to line 119-120.

The outcome section is well written.

Line 186: add that stratification is also done by gender.

Line 195-197: ‘After exclusion of individuals with missing data for at least one healthy behavior factor used for the HLI, the final study sample consisted of 7,382 participants for GNHIES 1990-92, 5,603 for GNHIES 1997-99 and 5,073 for GNHIES 2008-11’ were there any differences between participants with missing data and the included sample?

Line 207: ‘For age it was standardized to the German population structure as of 31 December 2010.’ To me this is unclear; since you will compare the prevalence of health behavior of people with a certain age (which is fixed since within each cohort you do not check changes over time) in 1990, 1997 and 2008. To what extend does age need to be standardized? Do you mean the results are weighted? And why standardized to 31 December 2010, though data has also been collected in 2011. Please clarify what you mean with this sentence.

Results

Authors present the data in the figure stratified by men and women, but why not by age groups or education level. Please elaborate or potentially add visual stratification in the supplement.

Were there any differences between 1990-92 and 1997-99 or between 1997-99 and 2008-11?

Line 235: was the proportion of men with high number of healthy behavior also significantly different.

Discussion

Line 271-274 (and line 376): add that trend also include data from 1997-99. Now the impression is that there were only two data collection points.

Line 287-289: “In line with this assumption, the cardiovascular disease mortality has declined in Germany in the last decades and mortality rates are expected to further decline in Germany until 2025 (35-37).” Do authors expect that this is due to a healthy lifestyle or better health care?

Line 306-307: can authors give a potential explanation for this observation? Different legislation?

Limitations

The response rate for GNHIES 2008-11 was much lower (42%) compared to the other two surveys (70% 1990-92 and 61% 1997-99). Please provide some explanation for the discrepancies.

A healthy diet is considered as daily fruit and vegetables intake. What about fibers, and no consumption of sweets and savory products, or soda?

Fruit and vegetable consumption was assessed by means of a self-administered food-frequency questionnaires. Intake was measured for the past 12 months for 1997-99, while it was measured for the past 4 weeks in 2008-11. Acknowledge the inconsistency and provide any expect differences?

Reviewer #2: This is an interesting study which highlight the importance of lifestyle change during the current epidemic of NCD in World.

Reviewer #3: Thank you for the opportunity to review your article titled, "Time trends in healthy lifestyle among adults in Germany: Results from three national health interview and examination surveys between 1990 and 2011".

Overall, the purpose of this study is interesting, and be well organized paper. However, several issues were still concerned.

Major points:

1) Selection bias: As the authors mentioned in line 370, selection bias (generalizability) would be more important in the present study. The authors must add values of response rates (not proportion of missing of HLI) not only for whole sample in each time point, but also response rates according to sex and age groups if possible. Additionally, the authors should consider generalizability by comparison with complete survey such as German census and the present dataset in each time point (e.g. comparison whether lower education level prevalence are same between German census and the present dataset).

2) Line 367: It was unclear what did the authors concerned as “bringing about inevitably changes in survey and assessment methods”. Please describe the detail which lifestyle assessment was not differed by period.

3) To understand what items mainly contributed to HLI ≥ 4, the authors should add results of cross table about proportion of each items of HLI by binary variable of HLI ≥ 4.

Minor points:

1) Tables: “Relative change 1990-2011” is inappropriate, because it would be 1990-92 vs. 2008-11.

2) Table 3: Tile “Proportions (%) of four or five healthy lifestyle indicators” is inappropriate. For example, “Proportions of adhering to healthy lifestyle combination” may be more appropriate.

3) Results of decrease trend of fruits and vegetables intake in Germany was definitely reported by the previous study (https://www.ncbi.nlm.nih.gov/pubmed/26934826). Reader would concern why fruits and vegetables intake was decreasing among German people. Have the authors checked the decrease trend according to education level?

4) Figures: Font sizes should be bigger.

6. PLOS authors have the option to publish the peer review history of their article (what does this mean?). If published, this will include your full peer review and any attached files.

Reviewer #1: Yes: Ilse Reinders

Reviewer #2: No

Reviewer #3: No

---

## [Author Response · Author response to Decision Letter 0]

21 Aug 2019

Review of MS number PONE-D-19-16785 Time trends in healthy lifestyle among adults in Germany: Results from three national health interview and examination surveys between 1990 and 2011

Response to the reviewer

Reviewer #1: 

Abstract

1) The abstract is well written and clear.

Methods: Please add e.g. sufficient/ xx amount to ‘Daily fruits and vegetables intake’.

Answer: For the comparison in this analysis, we could only use the frequency information on fruit and vegetable consumption, so daily amounts were not calculated. The indicator reflects therefore only a daily consumption of both fruits and vegetables. We reformulated this to “A ‘daily intake of both fruits and vegetables’…”.

2) Results: ‘Nevertheless, the gender difference in healthy lifestyle has increased’; Please clarify to which direction. Are men healthier compared to women?

Answer: Thank you for this comment. This sentence is unclear. We revised the sentence in the abstract as well as in the Results to: ‘The RC of increasing healthy lifestyle prevalence between 1990-92 and 2008-11 was stronger albeit on a higher level among women compared to men. Therefore, the gender difference in healthy lifestyle has increased, …’

3) Last sentence; start the sentence with referring to education, so it is directly clear that results for differences in education will be presented.

Answer: Thank you for this comment. We revised the sentence: “Among high educated men the prevalence of a healthy lifestyle increased between 1990-92 and 2008-11 from 10.6% to 16.3% (p=0.01) and among high educated women from 16.4% to 30.3% and also among medium educated women (10.9 to 16.6, p<0.01), but no significant increase in healthy lifestyle prevalence was observed among men with low and medium education and among women with low education level.”

4) Conclusions: four out of five vs. 4 or 5 mentioned in the method section.

Answer: “at least four out of five” is correct; we revised “4 or 5” to “at least four out of five” (line 37)

5) Ethics statement

Did the 1990-92 participants signed an informed consent?

Answer: No, participants did not sign an informed consent, but verbal consent was witnessed and formally recorded. We added the following sentence into the manuscript (line 107): “Verbal consent was witnessed and formally recorded in the surveys 1990-92 and 1997-1999.”

6) Introduction

Line 73-77: Can authors also relate these numbers to a higher prevalence of health related diseases or premature death?

Answer: We added the following sentences: “In addition, an unhealthy lifestyle with a lower number of healthy behavior factors has been linked to a higher risk of developing type 2 diabetes and cardiovascular diseases and to premature death (15-17)” (lines 70-72).

7) Materials and Methods

Line 92-94: Make clear that you compare prevalence data measured at 3 different time points in 3 different cohort. This is not always clear.

Answer: Thank you for this comment. Indeed, the analysis is based on three population-based, cross-sectional survey samples collected at three different time periods. We revised the following sentence in the section ‘Statistical methods’: ‘Age-standardized and weighted prevalence and 95% confidence intervals (CI) were calculated for each healthy behavior factor separately and for the HLI based on three cross-sectional survey samples collected at three different time periods (1990-92, 1997-99 and 2008-11)’ (page 11, lines 224-25).

8) Line 104: Mentioning the response rate is not in line with informing that response rates for the three surveys were published previously (line 97).

Answer: Thank you for this comment. We revised the sentence in line 97 to ‘Detailed information on the study designs and methods for the three surveys were published previously [1-4].’ (lines 95, 96), but did not change the sentence with information on response rate ‘The response rates were 70% for GNHIES 1990-92, 61% for GNHIES 1997-99 and 42% for GNHIES 2008-11 [4].’ 

9) Line 108 (as mentioned in the ethics statement): Why did participants from the 1997-99 wave did not sign an informed consent?

Answer: Participants in the 1997-1999 signed an informed written consent, but not in 1990-1992. We revised the sentence: “Verbal consent was witnessed and formally recorded in the surveys 1990-92 and 1997-1999. The GNHIES 1997-99 and GNHIES 2008-11 participants signed an informed written consent prior to participation” (page 6, lines 107, 108).

10) Line 119: ‘Smoking habits were assessed with the questions allowing a distinction between ‘current smokers’ and ‘others’’. This is a strict cut-off value. Please elaborate why this was so strict. Do authors have any information about former smoking?

Answer: Thank you for this comment. Information on former smoking was also available in the three surveys assessed with slightly different wordings. We added this information in line 122.

We selected the cut-off “current non-smoking” because it is an established WHO public health indicator. Furthermore, “current non-smoking” was used in healthy lifestyle scores of previous studies demonstrating strong associations between healthy lifestyle and health outcomes and mortality [5, 6]. We added a brief justification why we used this cut-off on page 9, lines 181-83.

11) Line 114 vs 121: clarify that a healthy diet is defined as fruit and vegetables intake.

Answer: Thank you for this comment. We inserted the sentence: “A daily consumption of both fruits and vegetables was used as an indicator for a healthy dietary pattern” (page 7, line 151).

12) Line 123-147: The explanation of alcohol consumption is too wordy which makes it difficult to follow. Recommend to rewrite/shorten or put to supplement.

Answer: Thank you for this comment. We put the explanation of alcohol consumption into an extra file (S1 File) and inserted the sentence “A detailed description of the alcohol assessment is given in the supplement (S1 File)” (line 149).

13) Line 150: Assessment of fruit and vegetables in 1990-92 is not mentioned to which time period is referred to.

Answer: Thank you for this advice. We changed the sentence into: “Information on frequency of consumption of fruit and vegetables was assessed in GNHIES 1990-92 with the question ‘How often do you consume these particular foods?’ without a specific reference period” (page 7, lines 153-156).

14) Line 148-168: Restructure, so all information is mentioned and provided in the same order.

Answer: We revised the text according to the reviewer’s suggestion. 

15) Line 170 is repetitive to line 119-120.

Answer: Thank you for this comment. See also answer on comment no. 10. We revised the text in both paragraphs. In the first paragraph we now focus on the measurement of smoking habits in the three surveys (line 122). In the second paragraph we focus on the variable definition and the justification of the cut-off used (lines 181-83).

16) The outcome section is well written.

Line 186: add that stratification is also done by gender.

Answer: We revised the sentence according to the reviewers suggestion: “We stratified the analyses by gender and used the following age strata….” (page 9, line 200).

17) Line 195-197: ‘After exclusion of individuals with missing data for at least one healthy behavior factor used for the HLI, the final study sample consisted of 7,382 participants for GNHIES 1990-92, 5,603 for GNHIES 1997-99 and 5,073 for GNHIES 2008-11’ were there any differences between participants with missing data and the included sample?

Answer: Thank you for this advice: in GNHIES 1990-92 98.9 % of participants were included in the study sample, in GNHIES 1997-99 96.2 % and in GNHIES 2008-11 94.4 %. We analysed the sample with and without missings and compared the deviation of the weighting factors. As there were no differences in these weighting factors between both samples, we used the complete case sample.

18) Line 207: ‘For age it was standardized to the German population structure as of 31 December 2010.’ To me this is unclear; since you will compare the prevalence of health behavior of people with a certain age (which is fixed since within each cohort you do not check changes over time) in 1990, 1997 and 2008. To what extend does age need to be standardized? Do you mean the results are weighted? And why standardized to 31 December 2010, though data has also been collected in 2011. Please clarify what you mean with this sentence.

Answer: Both is correct, the results are weighted with adaptive weights and age standardized. Adaptive weighting by age, sex, region and educational level was applied in a methodological consistent manner for all three surveys. In addition, the results were age-standardized to the German population structure as of 31 December 2010. The analysis is based on three population-based cross-sectional survey samples collected at three different time periods. As the demographic structure of the population in Germany has changed between 1990-92 and 2008-11 which is also reflected in the sample distributions, in terms that the population became slightly older over time, it is necessary to age-standardise the prevalence of healthy lifestyle across the three surveys according to a reference population. This allows investigating whether healthy lifestyle style prevalence has changed over time independently of the changing age structure of the population. We decided to use the population structure of the last survey (GNHIES 2008-2011) as the reference population to perform the age standardization. The population projection of the Federal Statistical Office of 31 December 2010 was also used for constructing the cross-sectional adaptive weighting factor for GNHIES 2008-11 because this date comes closed to the end of the data collection period of GNHIES 2008-11. We now clarify this in more detail in the ‘Statistical methods’ section (page 11, lines 220-22). 

19) Results

Authors present the data in the figure stratified by men and women, but why not by age groups or education level. Please elaborate or potentially add visual stratification in the supplement.

Answer: Figure 1 presents the trend of five health behaviour factors among men and women. Due to differences in some health behaviour factors we presented information by age groups for men and women separately in table S1 and S2. An additional stratification into education level was not valid due to small n per strata. Nevertheless, as suggested we added proportions of healthy behaviour factors according to educational level for the total sample in a Figure in the supplement (S1 Fig). Further stratification was not done as a detailed analysis is part of a next project which investigates absolute and relative social inequalities in health behaviour over time. We inserted the following sentence in the result section: “The increasing trend in sufficient physical exercise and no current risk drinking and the decreasing trend in daily fruits and vegetables intake over time can be observed in all educational groups albeit on a different level (S1 Figure). No significant changes in the prevalence of normal weight over time between different educated groups can be observed” (page 12, lines 248-252).

20) Were there any differences between 1990-92 and 1997-99 or between 1997-99 and 2008-11?

Answer: Yes, we added the following sentence: “More pronounced differences were observed for no current risk drinking with higher increases in prevalences between 1990-92 and 1997-99 compared to 1997-99 and 2008-11. The decrease in daily fruits and vegetables intake was more distinct between 1997-99 and 2008-11 in comparison to 1990-92 and 1997-99” (page 12, lines 237-40). 

21) Line 235: was the proportion of men with high number of healthy behavior also significantly different.

Answer: The written sentence is unclear and incomplete. We revised the sentence to “The proportions of men with none or only one healthy behavior factor have declined (relative change: -62.9% and -19.2%, both p < 0.0001), while the proportions of those with three (RC: +25.8%, p = 0.10), four (RC: +42.4%, p = 0.009) and five factors (RC: +66.7%, p < 0.0001) have increased” (page 13, line 258-61).

22) Discussion

Line 271-274 (and line 376): add that trend also include data from 1997-99. Now the impression is that there were only two data collection points.

Answer: We revised the sentence according to the reviewers suggestion: “…overall increased explicitly among adults in the period between 1990-92 and 1997-99 and further slightly between 1997-99 and 2008-11” (page 16, lines 312-13). 

23) Line 287-289: “In line with this assumption, the cardiovascular disease mortality has declined in Germany in the last decades and mortality rates are expected to further decline in Germany until 2025 (35-37).” Do authors expect that this is due to a healthy lifestyle or better health care?

Answer: We assume that it is due to both, a healthy lifestyle and better health care and added the following sentence: “This is on the one hand due to a better health care with improved detection and treatment of cardiovascular and metabolic diseases and on the other hand due to an improved health-related behaviour (28)” (page 17, lines 329-331).

24) Line 306-307: can authors give a potential explanation for this observation? Different legislation?

Answer: We insert the following two sentences with a potential explanation: “In Germany in the 1990s there was a discussion on a stronger legal alcohol limit when driving, which resulted in a revised law in 2001 (45). Furthermore, the national legislation has developed the banning of alcohol use at work within the labor protection law and employer agreements at the workplace (46)“ (page 18, lines 351-54).

25) Limitations

The response rate for GNHIES 2008-11 was much lower (42%) compared to the other two surveys (70% 1990-92 and 61% 1997-99). Please provide some explanation for the discrepancies.

Answer: There is evidence that response rates have declined in many studies in many countries in recent years and this is also the case in national health surveys [7, 8]. Nevertheless, declining response rates does not necessary mean that the samples are less representative. It is important that the composition of the realised sample is unbiased as possible. Selection bias can occur at different levels of recruitment (selection of sample points, selection of individuals, participation of individuals). 

In the current study weighting factors were applied to adjust for deviations of the sample compared to the general population structure to increase the generalizability of the results. We added a brief paragraph in the limitations section and elaborate more in detail on this issue (pages 21/22, lines 442-47).

26) A healthy diet is considered as daily fruit and vegetables intake. What about fibers, and no consumption of sweets and savory products, or soda?

Answer: Indeed, daily fruit and vegetables intake is only a rough proxy indicator of a healthy dietary behaviour. We added the following sentences in the “Limitations” section: “A consumption of both fruits and vegetables each day is a very rough indicator for a general healthy diet which may include much more aspects e.g. intake of highly processed foods, saturated fat, fiber, sugar intake. However, comprehensive information on the diet was not available in all three surveys”, (page 21, lines 435-38).

27) Fruit and vegetable consumption was assessed by means of a self-administered food-frequency questionnaires. Intake was measured for the past 12 months for 1997-99, while it was measured for the past 4 weeks in 2008-11. Acknowledge the inconsistency and provide any expect differences?

Answer: Thank you for this comment. We added the following sentences: “Consumption was assessed with self-administered food-frequency questionnaires with inconsistent reference periods and slightly different food items and answering categories for the three surveys. Although we tried to standardize this by condensing this information, this may have resulted in systematic differences in estimates for the surveys”, (page 21, lines 438-41). 

Reviewer #2: 

This is an interesting study which highlight the importance of lifestyle change during the current epidemic of NCD in World.

Reviewer #3: 

Thank you for the opportunity to review your article titled, "Time trends in healthy lifestyle among adults in Germany: Results from three national health interview and examination surveys between 1990 and 2011".

Overall, the purpose of this study is interesting, and be well organized paper. However, several issues were still concerned.

Major points:

28) Selection bias: As the authors mentioned in line 370, selection bias (generalizability) would be more important in the present study. The authors must add values of response rates (not proportion of missing of HLI) not only for whole sample in each time point, but also response rates according to sex and age groups if possible. Additionally, the authors should consider generalizability by comparison with complete survey such as German census and the present dataset in each time point (e.g. comparison whether lower education level prevalence are same between German census and the present dataset).

Answer: Thank you for this comment. See also answers on comments of Reviewer 1 no. 17 and 25. Unfortunately, the authors do not have information on response rates according to sex and age groups. Response rates in the German national health surveys were calculated as the number of participants divided by the number of invited sample members reduced by quality neutral losses. Nevertheless, all analyses were weighted. Weighting at the design level affects two probabilities: the selection of a particular sample point and selection of participants within the sample point. Weighting factors were applied to compensate for differences in willingness to participate with regard to the total German population structure according to age, gender, region and educational level. To compensate for a selection bias of the sample, the weighting factors were used to calculate prevalences. Using weighting factors, participants with lower educational level show higher weighting factors compared to participants with medium educational level. Therefore, prevalences in the survey can be generalized to the German population at a defined time point. 

From all participants in GNHIES 1990-92 98.9 % of them were included in the analysis, in GNHIES 1997-99 96.2 % and in GNHIES 2008-11 94.4 %. We analysed the sample with and without missings and compared the deviation of the weighting factors. As there were no differences in these weighting factors between both samples, we used the complete case sample.

29) Line 367: It was unclear what did the authors concerned as “bringing about inevitably changes in survey and assessment methods”. Please describe the detail which lifestyle assessment was not differed by period.

Answer: Unfortunately, the questions on smoking, physical activity, alcohol as well as fruit and vegetable consumption differed slightly in the wording between the surveys, e.g. in 1990-92 and 1997-99 smoking was assessed with the question “Have you ever smoked or do you smoke at the moment?” and in 2008-11 “Do you currently smoke – even if only occasionally?”. This was meant with “bringing about inevitably changes in survey and assessment methods“ and these changes should be considered when interpreting the results as we cannot rule out that this fact has been affected the results. According to the reviewer’s suggestion we added in the limitation section the following sentence to describe which assessment was not changed as requested:

“The initial physical exercise question has not changed across the three surveys, but the answer categories have been adapted. The initial smoking question was the same in the first two surveys but was adapted in the third survey. The BMI assessment method based on physical examination data on body weight and height has remained consistent across the three surveys (28).” (page 21, lines 431-35) .

30) To understand what items mainly contributed to HLI ≥ 4, the authors should add results of cross table about proportion of each items of HLI by binary variable of HLI ≥ 4.

Answer: Thank you for this comment, we added a table (top of page 14):

The proportion of sufficient physical exercise, daily fruits and daily vegetables, no current smoking, and normal weight as proportion of HLI increased over time. No current risk drinking did not show any changes over time (page 13, lines 277-79).

Minor points:

31) Tables: “Relative change 1990-2011” is inappropriate, because it would be 1990-92 vs. 2008-11.

Answer: Thank you for the comment! We changed the table description according to the reviewer’s suggestion.

32) Table 3: Title “Proportions (%) of four or five healthy lifestyle indicators” is inappropriate. For example, “Proportions of adhering to healthy lifestyle combination” may be more appropriate.

Answer: Thank you for this comment. We changed the title of the table into “Proportions (%) of adhering to a healthy lifestyle”. 

33) Results of decrease trend of fruits and vegetables intake in Germany was definitely reported by the previous study (https://www.ncbi.nlm.nih.gov/pubmed/26934826). Reader would concern why fruits and vegetables intake was decreasing among German people. Have the authors checked the decrease trend according to education level?

Answer: Thank you for the comment! According to the suggestion of Reviewer 1 (comment no. 19) we added an additional Figure in the supplement (S1 Figure) on proportions of healthy behaviour factors according to educational level. Furthermore, the following sentence was inserted in the text: “The increasing trend in sufficient physical exercise and no current risk drinking and the decreasing trend in daily fruits and vegetables intake over time can be observed in all educational groups albeit on a different level (S1 Figure)”, page 12, lines 248-51. Furthermore, we included the suggested reference on trends of fruits and vegetables from Gose et al. 2016 in the discussion on page 17, line 349.

34) Figures: Font sizes should be bigger.

Answer: Font sizes were adapted. 

References

1. Hoffmeister H, Bellach BM (1995) [Health of the Germans. A East-West comparison of health data] Robert Koch Institute, Berlin

2. Bellach BM, Knopf H, Thefeld W (1998) [The German Federal Health Survey 1997/98]. Gesundheitswesen 60 Suppl 2:S59-68

3. Scheidt-Nave C, Kamtsiuris P, Gosswald A et al. (2012) German health interview and examination survey for adults (DEGS) - design, objectives and implementation of the first data collection wave. BMC Public Health 12:730

4. Finger JD, Busch MA, Du Y et al. (2016) Time Trends in Cardiometabolic Risk Factors in Adults. Dtsch Arztebl International 113(42):712-719

5. Khaw K-T, Wareham N, Bingham S et al. (2008) Combined Impact of Health Behaviours and Mortality in Men and Women: The EPIC-Norfolk Prospective Population Study. PLOS Medicine 5(1):e12

6. Myint PK, Luben RN, Wareham NJ et al. (2009) Combined effect of health behaviours and risk of first ever stroke in 20,040 men and women over 11 years’ follow-up in Norfolk cohort of European Prospective Investigation of Cancer (EPIC Norfolk): prospective population study. Bmj 338:b349

7. Beullens K, Loosveldt G, Vandenplas C et al. (2018) Response Rates in the European Social Survey: Increasing, Decreasing, or a Matter of Fieldwork Efforts? Survey Methods: Insights from the Field DOI:10.13094/SMIF-2018-00003

8. Tolonen H, Helakorpi S, Talala K et al. (2006) 25-Year Trends and Socio-Demographic Differences in Response Rates: Finnish Adult Health Behaviour Survey. European Journal of Epidemiology 21(6):409-415

---

## [Editor Report · Decision Letter 1]

26 Aug 2019

Time trends in healthy lifestyle among adults in Germany: Results from three national health interview and examination surveys between 1990 and 2011

PONE-D-19-16785R1

Dear Dr. Finger,

We are pleased to inform you that your manuscript has been judged scientifically suitable for publication and will be formally accepted for publication once it complies with all outstanding technical requirements.

With kind regards,

David Meyre

Academic Editor

PLOS ONE
---

## [Editor Report · Acceptance letter]

29 Aug 2019

PONE-D-19-16785R1 

Time trends in healthy lifestyle among adults in Germany: Results from three national health interview and examination surveys between 1990 and 2011 

Dear Dr. Finger:

I am pleased to inform you that your manuscript has been deemed suitable for publication in PLOS ONE. Congratulations! Your manuscript is now with our production department. 

With kind regards,

on behalf of

Dr David Meyre 

Academic Editor

PLOS ONE